# C-Phycocyanin and Phycocyanobilin as Remyelination Therapies for Enhancing Recovery in Multiple Sclerosis and Ischemic Stroke: A Preclinical Perspective

**DOI:** 10.3390/bs8010015

**Published:** 2018-01-18

**Authors:** Giselle Pentón-Rol, Javier Marín-Prida, Viviana Falcón-Cama

**Affiliations:** 1Center for Genetic Engineering and Biotechnology (CIGB), Ave. 31 e/158 y 190, Cubanacan, P.O. Box 6162, Playa, Havana 10600, Cuba; viviana.falcon@cigb.edu.cu; 2Center for Research and Biological Evaluations (CEIEB), Institute of Pharmacy and Food, University of Havana, Ave. 23 e/214 y 222, La Lisa, PO Box 430, Havana 13600, Cuba; jmarin@infomed.sld.cu

**Keywords:** C-phycocyanin, Phycocyanobilin, remyelination, Multiple Sclerosis, stroke, recovery

## Abstract

Myelin loss has a crucial impact on behavior disabilities associated to Multiple Sclerosis (MS) and Ischemic Stroke (IS). Although several MS therapies are approved, none of them promote remyelination in patients, limiting their ability for chronic recovery. With no available therapeutic options, enhanced demyelination in stroke survivors is correlated with a poorer behavioral recovery. Here, we show the experimental findings of our group and others supporting the remyelinating effects of C-Phycocyanin (C-PC), the main biliprotein of *Spirulina platensis* and its linked tetrapyrrole Phycocyanobilin (PCB), in models of these illnesses. C-PC promoted white matter regeneration in rats and mice affected by experimental autoimmune encephalomyelitis. Electron microscopy analysis in cerebral cortex from ischemic rats revealed a potent remyelinating action of PCB treatment after stroke. Among others biological processes, we discussed the role of regulatory T cell induction, the control of oxidative stress and pro-inflammatory mediators, gene expression modulation and COX-2 inhibition as potential mechanisms involved in the C-PC and PCB effects on the recruitment, differentiation and maturation of oligodendrocyte precursor cells in demyelinated lesions. The assembled evidence supports the implementation of clinical trials to demonstrate the recovery effects of C-PC and PCB in these diseases.

## 1. Introduction

Myelin is a lipid-rich structure that wraps the axons in a multilayered organization, as cellular membrane extensions of oligodendrocytes (ODs) or Schwann cells in the central or peripheral nervous systems, respectively. These myelin sheaths have a fundamental role in the nervous system, allowing not only the rapid axonal conduction of electrical impulses with minimal energy across considerable distances but also the maintenance of axonal integrity and health through trophic and metabolic support [1]. The crucial importance of myelin for the higher complex functions of mammalian nervous systems can be noticed from their relative composition, as for example, approximately 40% of the human brain consists of white matter, in which 50–60% of dry weight contains myelin as the main component [2]. The impact of myelin loss, defined as demyelination and the formation of new functional myelin sheaths or remyelination, has received increasing attention in recent years and it is now recognized as a major factor involved in the pathogenesis of neurodegenerative diseases, especially Multiple Sclerosis (MS) and Ischemic Stroke (IS) [3].

Behavioral deficits are the most important sequelae in stroke survivors. Although motor dysfunction and its compensatory mechanisms is one of the most widely studied neurological outcomes of cerebral ischemia [4], other serious behavioral disturbances are also involved, such as those involved in executive functions (e.g., planning, initiation, staying mindful of task objectives, prioritizing goals, sequencing activities for specific goals, inhibiting attention to irrelevant/distracting stimuli), aphasia (dysfunction in the expression or interpretation of spoken symbolic language), attention, memory, emotion and sensory processing [5]. The importance of myelin disruption in post-stroke sensory and motor deficits has been highlighted in both clinical and animal studies [6]. In a previous clinical assessment of 42 stroke patients enrolled in 6 neuroprotection trials, Ho et al. 2005 found that white matter occupies a median of 49% of the infarct volume, while 95% of all infarcts involved some regions of the white matter tracts [7]. In a recent study, the magnetic resonance imaging analysis of the ipsilesional corticospinal tract and corpus callosum of 18 survivors suffering from hand motor deficits, in the subacute or chronic stages after subcortical IS, showed that the destruction of myelin sheaths and the associated axonal damage correlated with a poorer motor performance. The authors concluded that degeneration of transcallosal fibers connecting higher order sensorimotor regions may constitute a relevant factor influencing cortical reorganization and motor outcome after subcortical stroke [8]. These findings provide a strong rationale for investigating the effects of myelin loss/formation as well as for developing myelin protective or repairing therapies for IS.

Demyelination is a central pathophysiological hallmark in MS, a chronic inflammatory and neurodegenerative disease of the central nervous system (CNS) that is thought to have an autoimmune etiology [9]. MS patients suffer discrete episodes (“attacks” or “relapses”) of neurological impairments. In the relapsing-remitting stage of the disease (RRMS), patients remain stable between these attacks. However, around 80% of those patients that do not receive any disease-modifying treatment evolve into the secondary progressive phase of the illness (SPMS) within 20 years after the onset [10]. In this subset of patients, the inflammatory-demyelinating injury is disseminated and they experience an insidious worsening of clinical symptoms and the accumulation of neurologic disability unrelated to any acute attacks. A small percentage of patients (10–20%) have a clinical course of primary progressive (PP) MS, in which they only experience insidious worsening and never have acute attacks. The clinical symptoms of MS include numbness, tingling, motor weakness, vision loss, gait impairment, incoordination, imbalance, spasticity, bladder issues, headache and psychological dysfunction such as cognitive problems (slowed information processing, executive dysfunction, impairment of long-term verbal and visual memory), fatigue and depression [11]. These symptoms are the major drivers in the worsening of MS patient’s quality of life, leading to a negative impact on their social, professional and family activities [12]. In an Australian epidemiological study involving 1329 MS patients, Simmons et al. 2010 showed that between 2003 and 2007, despite being a period of relative national economic prosperity, 56% of them had lost their jobs due to MS, primarily caused by the ineffective management of MS neurological symptoms at the workplace [13]. Until 2016, all disease-modifying MS therapies were approved for relapsing forms of the disease, acting primarily by modulating or suppressing the immune system to reduce relapse rates and magnetic resonance imaging measures of inflammation but these medications are unable to effectively reverse the chronic (progressive) phase of the disease [14]. More recently, after three positive clinical trials [15,16], Ocrelizumab (OcrevusTM)—a humanized anti-CD20 monoclonal antibody—was approved in March 2017 by the U.S. Food and Drug Administration for the treatment of both the relapsing forms of MS and PPMS [17]. However, none of these drugs directly improve the remyelination process and therefore their ability to regenerate and promote the chronic repair of MS lesions is limited [18]. It is therefore imperative to design novel therapeutic strategies aimed at promoting remyelination in the CNS, preferably at the early stages following the onset of MS, which may stop the decline toward the progressive phase of the disease. During this progressive phase of MS, acute inflammatory relapses are scarce but there is increasing loss of neuronal function and extensive demyelination and thus, the stimulation of endogenous remyelination would potentially have the greatest clinical impact [19].

This review provides a summary of preclinical evidence supporting the application of two naturally occurring compounds, C-Phycocyanin and Phycocyanobilin, as potential agents for promoting neural tissue regeneration in MS and IS, primarily focused on their remyelination inducing properties. We will present, discuss and comment results from our group and other authors demonstrating the beneficial actions of these compounds in experimental models of MS and IS as remyelination inducing agents, which support their medical application for these diseases.

## 2. C-Phycocyanin and Phycocyanobilin

C-Phycocyanin (C-PC), the main biliprotein of the cyanobacteria *Spirulina platensis* [20] and its associated open-chain tetrapyrrole chromophore named Phycocyanobilin (PCB) [21,22,23] exhibit an array of biological properties such as antioxidant and anti-inflammatory characteristics, in addition to immunomodulatory actions [24]. For more than a decade, our group has been exploring different experimental approaches to identify the pharmacological properties of C-PC and PCB, including their neuroprotective and neurorestorative activities. These are linked not only to their well-known antioxidant, free radical scavenging and anti-inflammatory properties [25,26] but also to the modulation of the expression of several genes associated with other biological processes.

## 3. Mechanisms of Remyelination: An Overview

Remyelination is a restorative phenomenon in which new myelin sheaths are formed on demyelinated axons following an injury to the nervous system. However, this regenerative process is limited in MS, mainly owing to the failure of the oligodendrocyte precursor cells’ (OPCs) recruitment into the lesions and the inability of these cells to differentiate into myelinating ODs [27,28]. OPCs are present throughout both gray and white matter in the CNS and have “stem cell-like” properties such as multipotency and self-renewal [29]. In response to demyelination, OPCs must proliferate and migrate to the lesion site [30] where they differentiate into mature ODs, extending processes to remyelinate denuded axons [31] and consequently, saltatory conduction is restored and axons are protected from further degeneration [32]. In some paradigms, while axons are not fully protected, their degeneration is substantially delayed, where motor deficits do not re-appear until much later in time [33]. Several factors may perturb any of these stages, or even the disruption of one phase may affect the others [34]. Using autopsy samples from MS patients, OPCs have been found in chronically demyelinated lesions, with a density similar to those in the developing rodent brain but unable to sheathe the injured dystrophic and swelled axons [35]. Even in OPCs that differentiate, contact and enwrap axons, the final maturation into compact myelin may be interrupted [36]. Similarly, the susceptibility of white matter to stroke has been associated to the vulnerability of OPCs to a diversity of insults, including oxidative stress, trophic factor deprivation, excitotoxicity and the activation of apoptotic pathways [37]. This suggests that cellular and molecular interactions in the microenvironment of demyelinated lesions, including axons, astrocytes, immune cells, microglia, OPCs and premyelinating ODs, are limiting factors for effective remyelination [38]. Thus, at least two general strategies may be envisioned in order to enhance remyelination either in MS or IS: to promote the recruitment and differentiation of OPCs into the lesion areas; and to overcome inhibitors of OPCs maturation present within these lesions [39,40].

## 4. C-PC Remyelinating Actions in MS Models

Our group has previously reported remyelinating actions of C-PC in different models of MS. In Lewis rats with experimental autoimmune encephalomyelitis (EAE), a MS model in which paralysis is caused by an immune response against CNS myelin protein antigens [41], C-PC at 25 mg/Kg/day significantly alleviated the clinical progression of the disease when the animals were treated intraperitoneally starting either 12 days before (prophylactic regimen) or from day 0 to 12 after immunization (early therapeutic regimen). It is noteworthy that none of the rats in the prophylactic group developed any disease symptoms even after 24 days post-immunization (end of the study), indicating that this treatment regimen completely abrogates disease onset. Our data also showed that the early C-PC therapeutic schedule not only significantly decreased the maximal clinical score of the diseased rats but also accelerated the end of the clinical symptoms by 5 days in comparison to the EAE vehicle-treated group. We also performed transmission electron microscopy analysis in brain samples taken at the end of the study period. Compact and dense myelin and no signs of axonal damage were observed in biopsies from the non-EAE group. In contrast, disease induction in the EAE rats was accompanied by loosened, wobbly and unfastened myelin and axon injury reflected by the mitochondrial dilatation observed within most axons. Rats treated with C-PC had compressed, solid and squashed myelin and no signs of axonal breakdown, a pattern comparable to the control animals [42]. Quantitative evidence for C-PC-induced remyelination reported by our group in this previous study is shown for the first time here by assessing the *g*-ratio (ratio of axon diameter to myelinated axon diameter). Panels A–C from Figure 1 exhibit representative myelin-axon structures from the study animal groups, as described above, while Figure 1D shows the determination of the *g*-ratio. Normal myelinated axons have a *g*-ratio between 0.6 and 0.8, whereas a demyelinated axon typically has a *g*-ratio between 0.8 and 1, a value that indicates complete myelin loss [43]. Accordingly, the *g*-ratio for EAE vehicle-treated animals was significantly higher as compared to non-diseased rats (mean ± S.E.M.: 0.88 ± 0.004 and 0.66 ± 0.017, respectively). Notably, the early therapeutic C-PC treatment was able to restore the *g*-ratio level (0.67 ± 0.007) to those observed in the normal group. This evidence confirmed that remyelination took place in the brain of the EAE rats due to the C-PC treatment, an effect that was accompanied by a significant improvement in the levels of redox biomarkers, such as malondialdehyde, lipid peroxidation potential, total organoperoxides, ferric reducing ability and advanced products of protein oxidation, either in the serum or in the brain homogenates of diseased rats. Our results may have a relevant clinical impact since cortical demyelination and retrograde neurodegeneration in the brain of MS patients have been associated with oxidative injury/meningeal inflammation and white matter demyelinated lesions/axonal loss, respectively [44].

Recently, by using biopsies of MS patients and an anti-myelin antibody-dependent experimental model, Lagumersindez-Denis et al. 2017 have shown that the mechanisms involved in this phenomenon are also topographically distributed; with lesser restrictive inflammatory requirements for subpial cortical demyelination than those occurring in perivascular areas. Demyelination in these cortical lesion types was dependent on the presence of pathogenic demyelinating antibodies and inflammatory monocytes but the perivascular cortical demyelination also included activated encephalitogenic T cells and NK cells [45].

In the above-mentioned study performed by our group [42], we also proposed a new focus of neuroprotection aiming to reestablish the effector/regulator balance of the immune response once it has been disrupted, a situation that arises in human autoimmune diseases such as MS. We studied the modulation of the T cells subset labeled with CD4+CD25highFoxp3+ after the *in vitro* treatment with C-PC of peripheral blood mononuclear cells (PBMC) either from MS patients or from healthy humans. Our results demonstrated a significant up-regulation of these regulatory T cells (Treg) markers due to the C-PC treatment in PBMC from MS patients, with no increase in the expression levels of CD69, an early marker of T cells activation [46]. Rodent EAE studies have demonstrated that disease development is dependent of the escape of effector auto-reactive T cells from the control of regulatory immune mechanisms that keep autoimmunity in check. The main mechanism is based on naturally occurring Treg cells (CD4+CD25+ T regulatory subset originated in the thymus). Acute depletion of Treg exacerbates EAE severity accompanied by increased pro-inflammatory cytokine levels, proliferation of effector T cells and their improved motility (as evidenced by a longer vascular stationary phase) in the CNS [47]. In MS patients with stable disease, Treg are significantly increased in the peripheral blood compared to those with the acute (clinical relapses) disease, thus suggesting a pivotal role of Treg in controlling MS relapses and probably disease progression [48]. In line with our results, Dombrowski et al. 2017 recently demonstrated that Treg promotes ODs differentiation and remyelination. Treg-deficient mice exhibited substantially impaired remyelination and ODs differentiation, which was rescued by adoptive transfer of Treg. Furthermore, Treg directly promoted OPCs differentiation and myelination *in vitro* [49]. Thus, by inducing Treg and through their mediation, C-PC can promote the crucial stages necessary for functional remyelination in MS.

Other biological properties of C-PC could also mediate its remyelination effects. Both cyclooxygenases (COX) isoforms, the constitutive COX-1 and the inducible COX-2, oxidize arachidonic acid via the short-lived hydroperoxyl-containing intermediate prostaglandin (PG) G_2_ to PG H_2_, which is then used by PG synthases for generating PGs D_2_, E_2_, F_1α_, F_2_, I_2_ (prostacyclin) and by thromboxane (TX) synthase for producing TX A_2_ and B_2_ [50]. COX-2 is upregulated in the CNS under pathophysiological conditions and has been associated with neurodegenerative mechanisms [51]. COX-2 expression is increased in ODs present in chronic active lesions of brain biopsies from MS patients and from spinal cord lesions of mice with Theiler’s virus induced demyelination [52]. A recent study has also shown that PGE_2_ and its EP1 receptor in OPCs directly block the maturation of these cells *in vitro*, an effect that is attenuated by EP1-specific inhibitors, or by the genetic deficiency of EP1. Furthermore, *in vivo* inhibition of COX-2 with nimesulide rescues IL-1β-induced hypomyelination in the brains of neonatal mice [53]. OPCs cultures prepared from mice brains have also shown significant increases in PGE_2_ and death levels when they are incubated with kainic acid, a compound that mediates excitotoxic insults. The kainic acid-induced deleterious effects are significantly diminished when OPCs are treated with CAY10404, a COX-2 specific inhibitor, mediated by decreasing the PGE_2_ production and the subsequent inhibition of PGE_2_ receptor activity [54]. Hence, this data supports the negative impact of COX-2 in OPCs survival and maturation. C-PC is also a selective COX-2 inhibitor, with an IC_50_ for this enzyme that is much lower and with a selectivity index (SI = IC_50_ COX-2/IC_50_ COX-1) much greater than the first generation of selective COX-2 inhibitors such as celecoxib and rofecoxib [55] and comparable to the second generation of this type of drugs, such as valdecoxib and etoricoxib [56]. Given that these compounds have several adverse effects limiting their clinical use [57], the medical application of C-PC as a selective COX-2 inhibitor may be a safer strategy [58] for promoting OPCs survival and therefore their proliferation in CNS demyelinating diseases such as MS.

Uncontrolled production of reactive oxygen species—such as peroxynitrite—is another factor that may also affect the myelogenesis process. Peroxynitrite is formed by the reaction of nitric oxide with the radical superoxide anion (O_2_^•−^). It may directly react with thiols and its decomposition radical products may oxidize lipids, proteins and DNA, eventually leading to cellular death [59]. It has been shown that induction of nNOS (neuronal isoform of nitric oxide synthase) in lipopolysaccharide (LPS)-stimulated OPCs *in vitro* results in increased levels of peroxynitrite and tyrosine nitration of proteins, as well as in a significant decrease of OPCs survival. The rise in peroxynitrite and tyrosine nitration accumulation, along with the demyelination evidenced by Luxol Fast Blue- Periodic Acid Schiff (LFB-PAS) histology and myelin basic protein (MBP) immunostaining, was also present in the *corpus callosum* of rats after the stereotaxic injection of LPS [60].This LPS model might represent the type III oligodendrogliopathy that is seen in a subset of MS patients with a lack of inflammatory cytokines and mild T cells infiltration in hypoxia-like lesions [61]. Peroxynitrite has also been critically involved in the excitotoxic-induced death of OPCs [62] and in the microglia-mediated damage of these cells in cerebellar organotypic slice cultures [63]. Previous studies from other authors have demonstrated that C-PC is an efficient scavenger of peroxynitrite [64] and significantly inhibits the LPS-induced nitrite production and iNOS (inducible isoform) protein expression in LPS-stimulated RAW264.7 macrophages [65]. Furthermore, C-PC was also able to significantly reduce the LPS-induced up-regulation of iNOS, along with a decrease in the expression of COX-2 and proinflammatory cytokines TNF-α and IL-6 mRNAs in LPS-stimulated BV-2 microglial cells [66]. Taken together, this evidence strongly points into remyelination actions of C-PC in MS by enabling OPCs survival, proliferation and maturation, through a versatile arsenal of biological abilities.

In a more recent study, our group described the remyelinating and ameliorating actions of C-PC in the MOG_35–55_ mice model of EAE [67]. Three doses of this drug (2, 4 or 8 mg/Kg) were intraperitoneally administered once a day at disease onset (therapeutic regimen) and extended for 15 days. We found a significant reduction in the inflammatory infiltration and axonal damage in the white matter areas of mice spinal cords treated with C-PC at 4 and 8 mg/Kg. We also observed a significant decrease in the levels of Mac-3 activated macrophages/microglia and CD3-positive T cells specifically present in the white matter lesions, as assessed by immunohistochemistry. In this study, C-PC was also able to protect the neuronal structures from deleterious processes, evidenced by a reduction in the density of APP-positive labeling in the white matter lesions, a marker indicative of acute axonal damage. By using LFB-PAS staining, we observed significant remyelination in the spinal cords of diseased mice treated with C-PC. As a follow-up assessment, here we report for the first time, electron microscopy analysis of the spinal cords of mice from this study thereby revealing more accurate details regarding the C-PC effects on myelin regeneration. Figure 2 confirms that EAE caused a massive damage to myelin structure (Figure 2B), while the treatment with C-PC at 8 mg/Kg completely restored the compact structure characteristic of functional myelin (Figure 2C), similar to that of normal mice (Figure 2A). This fact was also evidenced quantitatively by the C-PC-induced restoration of the *g*-ratio levels to those reached by the non-diseased mice. The *g*-ratio values for control (non-immunized), EAE-vehicle and EAE-C-PC 8 mg/Kg treated mice were, respectively (mean ± S.E.M.): 0.67 ± 0.029, 0.87 ± 0.007 and 0.70 ± 0.023. Our findings may also suggest that remyelination could play a role in the immediate axonal recuperation after the demyelinating insult [68].

In the latter work, we also demonstrated that C-PC restored the oxidative stress balance in EAE animals. C-PC at 8 mg/Kg significantly reduced the blood levels of malondialdehyde, the lipid peroxidation potential and the CAT/SOD index as compared with the EAE vehicle-treated group. The first two parameters reflect oxidative damage to lipids, which compromise the integrity and function of biological membranes [69], thus preventing lipid peroxidation which is of crucial importance given the lipid-rich nature of the myelin structure. In the same study, by using microarray analysis in brain samples taken at day 18 post-immunization, when EAE vehicle-treated mice reached the maximal neurological deterioration but the disease was already alleviated by C-PC, we identified a subset of 918 genes differentially regulated by this compound at the highest dose (536 genes up- and 382 genes down-regulated). This analysis helps shed light on the molecular mediators of the beneficial effects of C-PC in EAE. Interestingly, a set of genes associated with remyelination were up-regulated by C-PC (and confirmed by qPCR), such as *Mal*, *Mog* and *Mobp*, which are structural components of the myelin sheath, the transcription factors *Olig1* (present exclusively in ODs), *Nkx6-2* and *Nkx2-2*, all of them involved in key steps of OPCs differentiation and maturation [70,71,72]. Interestingly, the proteolipid Mal is a main component of lipid rafts in myelinating cells and is associated with glycosphingolipids. These Mal–glycosphingolipid interactions are believed to result in the formation of protein–lipid microdomains in myelin sheaths [73]. Schaeren-Wiemers et al. 2004, by using electron microscopy, have described that *Mal*-deficient mice contain conspicuous cytoplasmic inclusions within compact myelin and a large number of abnormal paranodes, with everted paranodal loops projecting away instead of contacting the axon. These altered axon–glia interactions result in molecular disorganization of nodes of Ranvier in *Mal* knock-out mice, evidenced by a drastic reduction in the expression of crucial proteins in the paranodal regions, such as Caspr, a glial-axon junction protein and also in juxtaparanodal areas of the axonal membrane, as evidenced by reduced levels of the Kv1.2 potassium channel [74]. Taken together, these findings may indicate that C-PC induces a gene expression profile that not only promotes the OPCs differentiation, maturation and axonal ensheathment but also the regeneration of normal functioning nodes of Ranvier in remyelinated axons of EAE mice.

On the other hand, in a recent work from our group, we have also observed that PCB treatment ameliorates the disease progression in a mice model of EAE, with positive actions on the resolution of inflammation and myelination (unpublished data). As C-PC is proteolytically degraded to PCB and PCB-linked small peptides once it is administered *in vivo* [75], our observations may indicate that this tetrapyrrolic compound could be mainly responsible for the C-PC pharmacological actions described in the MS models.

## 5. PCB Remyelinating Effects in Animal Models of Focal and Global Cerebral Ischemia

As stated above, white matter injury is also critically involved in neurological dysfunction after cerebral ischemia. Novel therapies that are able to promote the reformation and/or protection of the myelin against ischemic insults may be an effective way of enhancing regeneration and recovery in stroke survivors. However, translational gap has been huge in the IS research and a new concept for the ischemic penumbra as a critical transition zone between injury and repair has been proposed in order to speed up the bench-to-bedside application of effective interventions [76]. Our group has studied the effects of the PCB treatment in a rat cerebral hypoperfusion model induced by permanent bilateral common carotid arteries occlusion (BCCAo). Between 1–3 days following this surgical procedure, global cerebral blood flow is reduced to levels similar to those reached by the ischemic penumbra area after a focal IS, thereby allowing us to study the effects of potential cerebroprotectants in biological processes that are relevant for penumbra tissue preservation [77]. By using the BCCAo rat model, we performed microarray (Affymetrix GeneChipTM Rat Gene ST 1.1) and qPCR analyses, which revealed that PCB—administered intraperitoneally at cumulative doses of 47 or 213 μg/Kg for 30 min, 1, 3 and 6 h after surgery—promotes a positive modulation of immunological processes due to the effective regulation of anti-inflammatory and regulatory gene expressions [78]. A total of 190 genes were modulated by PCB at the highest dose (93 up- and 97 down-regulated genes, q-value < 1.05, fold change > 1.5), in the rat anterior cerebral cortex 24 h after the BCCAo surgery. qPCR assays showed that, at the same time point, this drug induced a dose-dependent positive regulation of 14 genes (*IFN-γ*, *IL-6*, *CD74*, *CCL12*, *IL-17A*, *Foxp3*, *IL-4*, *TGF-β*, *Mal*, *NADH dehydrogenase*, *Bcl-2a1*, *Baiap2*, *C/EBPβ and Gadd45g*) in the olfactory bulb, the anterior cerebral cortex and the hippocampus. While in the striatum, we observed the PCB-induced positive modulation of the mRNA expression of another 5 genes (*CXCL2*, *ICAM-1*, *IL-1β*, *TNF-α* and *VEGFA*) similar to the quantitative changes of their polypeptidic products in serum as assessed by Bio-Plex^®^.

Several of these genes modulated by PCB in this study performed by our group, have a direct impact on the remyelination process. For example, the significant increase in *Foxp3* mRNA levels in the BCCAo PCB-treated group may indicate a putative contribution of Treg cells to cerebroprotection under a condition of mild cerebral blood flow disruption, occurring in the ischemic penumbra, as previously reported by Liesz et al. 2009 [79] but not in cases of larger ischemic damage [80]. In three hypoperfused cerebral regions of rats treated with the vehicle, we also detected a significant decrease in the *TGF-β* expression, which was counteracted by the PCB treatment, increasing the mRNA of this gene even to levels significantly higher than the sham (control) animals. Treg cells may serve as a source of this anti-inflammatory cytokine [81], which has been reported to exert a neuroprotective role against ischemic stroke [82]. TGF-β also has a positive effect on remyelination. When TGF-β is overexpressed, oligodendrogenesis and subcortical white matter myelination are enhanced; in contrast, when its receptor (TGFβ-RII) is deleted in OPCs, the development of these cells into mature myelinating ODs is prevented, leading to hypomyelination in mice [83].On the other hand, the product of another gene, *IL-17A*, which was down-regulated by the PCB treatment in the BCCAo rat model, adversely affects not only the functioning of ODs but also of microglia, astrocytes, neurons, neural precursor cells and endothelial cells in MS [84]. Furthermore, mice deficient in *IL-17A* and its receptor (IL-17RC) have reduced demyelination in response to cuprizone feeding [85], while the transfer of Th17 CD4(+) T cells into cuprizone-fed mice impaired spontaneous remyelination [86]. Another PCB down-regulated gene, *IL-1β*, inhibits OPCs migration through its IL-1R1receptor. When the IL-1 receptor antagonist (IL-1Ra) is delivered at an early disease stage, as well as when IL-1R1 is silenced, the downregulation of MBP is rescued and the remyelination is improved at later stages after chronic cerebral hypoperfusion in mice [87]. TNF-α, a harmful cytokine also limited by the PCB treatment, has been shown to arrest OPCs maturation [88]. Thus, by inducing the Treg phenotype (Foxp3+), increasing the *TGF-β* expression and avoiding the rise of *IL-17A*, *IL-1β* and *TNF-α* genes expression in rat hypoperfused brains, PCB may lead into myelin reformation in the ischemic penumbra areas at very early stages after stroke, preventing the loss of this structure and eventually contributing to behavioral recovery in the chronic stages of the disease.

Additionally, we also detected an increase in the *Mal* gene expression induced by PCB treatment in BCCAo rats. As mentioned before, *Mal* expression has been strongly correlated with enhanced myelin formation and maturation into a compact functional structure by oligodendrocytes [73], thus potentially preventing the BCCAo characteristic white matter injury [89]. Furthermore, PCB also positively modulated the expression of *NADH dehydrogenase*, *Bcl-2a1*, *Gadd45g*, *Baiap2* and *VEGFA* genes, offering alternative potential mechanisms of cerebroprotection against brain hypoperfusion mediated by energetic metabolism, anti-apoptosis, synaptic plasticity and angiogenesis. In this sense, it has recently been shown that complex I (as detected by one of its subunits, ND4L), along with other members of the mitochondrial electron transport chain, such as COX IV (subunit of Complex IV) and the β subunit of F_1_F_o_-ATP synthase, are expressed in the CNS myelin sheaths, which confirms the trophic support of myelin to the axon [90]. In our study, we observed that PCB treatment up-regulated one component of the respiratory Complex I in BCCAo rats, the NADH dehydrogenase (ubiquinone) 1 beta subcomplex 2, thus suggesting its role in protecting axonal degeneration by energy supply coming from the newly formed myelin after cerebral hypoperfusion.

Our study also confirmed the antioxidant abilities of the PCB treatment. In several tissue compartments of BCCAo rats, such as serum, cerebral cortex, striatum and hippocampus, the administration of this compound produced a dose-dependent decrease of malondialdehyde and lipid peroxidation potential levels. These results highlight the importance of PCB in preventing oxidative damage to myelin, which contain, as mentioned above, a lipid rich composition (the estimated lipid to protein ratio in myelin is 186:1) [91]. PCB has been shown to be a more efficient scavenger than C-PC for several reactive species, such as peroxynitrite [64] and peroxyl radicals [92]. Thus, PCB has the capacity of inhibiting the formation of the dangerous hydroxyl radical (^•^OH) derived from this nitrogen reactive species and also the propagation of lipid radicals, with the consequent inhibition of lipid peroxidation in the myelin membrane [93].

More recently, in a rat model of focal cerebral ischemia induced by endothelin-1 (ET-1), our group have confirmed the remyelination actions of the PCB treatment. The drug was intraperitoneally administered at a cumulative dose of 200 µg/Kg for 30 min, 1, 3 and 6 h after the stereotaxic injection of ET-1 to the piriform cerebral cortex of Wistar rats, a procedure that renders a reproducible brain infarct in the middle cerebral artery territory covering ~40% of hemispheric cerebral volume, as previously described [94]. Cerebral cortex biopsies were taken at 24 h post-stroke and used for transmission electron microscopy analysis. As shown in Figure 3, the brain cortex of sham animals had compact and dense myelin and no signs of axonal damage (Figure 3A). Thinner and broken myelin was observed in the ET-1 vehicle-treated animals (Figure 3B). In contrast, ischemic rats treated with PCB showed a structural pattern of myelin similar to the sham group (Figure 3C). The quantitative analysis of myelination confirms the remyelination effects of PCB (Figure 3D), with *g*-ratios for the sham, ischemic-vehicle and ischemic-PCB treated rats of 0.65 ± 0.01, 0.82 ± 0.003 and 0.64 ± 0.008, respectively (mean ± S.E.M.). This evidence strongly points to a reparative role of PCB in the brain myelin structure after an ischemic insult.

Given the neuroprotective effects of C-PC in several IS models, its versatile antioxidant, immunomodulatory and gene/intracellular signaling regulatory abilities [95] and the myelinating actions here described for its derivative PCB in a stroke model, it is reasonable to hypothesize that C-PC would also be able to promote myelin reformation and enhance CNS repair after an ischemic injury. In line with this, our group previously observed the neuroprotective effects of the C-PC therapeutic intervention in a global cerebral ischemia/reperfusion injury in gerbils [96] and in transient focal ET-1-induced brain ischemia in rats (manuscript in preparation) [97]. Its anti-apoptotic actions in brain cells observed either *in vitro*, e.g., in SH-SY5Y neuronal lineage [98,99] and in primary cerebellar neurons [100], or *in vivo*, e.g., ischemic rat retina [99], kainic acid-induced seizures [101] and tinnitus-related excitotoxicity in the cochlea and inferior colliculus [102], also suggest its potential role for inducing OPCs survival and proliferation under hypoxic/ischemic conditions. Recently, Min et al. 2015 demonstrated that C-PC administered by the intranasal route at single doses of 67, 134, or 335 μg/Kg 1 h post-stroke, was able to significantly reduce the infarct volume and the neurological deficit assessed 48 h after the injury, in rats subjected to the transient occlusion (60 min) of the middle cerebral artery. Notably, the administration of 134 μg/Kg C-PC at 3, 6, or 9 h post-surgery significantly reduced the mean infarct volumes to 44.7 ± 9.5%, 56.3 ± 7.0% and 72.5 ± 3.1% in relation to the ischemic vehicle-treated group (considered as 100% infarct), respectively [103]. Nasal administration for delivering drugs to the brain has received special attention because it is less invasive than injections into veins, it can circumvent the hepatic first-pass elimination and it may also surpass the obstacles of the blood brain barrier [104]. Furthermore, the relatively wide therapeutic window of 9 h reported in this study for the nasal delivery of C-PC and possibly also for its derived PCB linked tetrapyrrole, strongly support its clinical application in the IS area, since the time window for post-stroke therapy has been recognized as a critical parameter in achieving clinical success [105].

Additionally, Min et al. 2015 also showed that C-PC protects 3D cultures of primary astrocytes (developed in polycaprolactone nanofibrous mats as scaffold) against H_2_O_2_-induced oxidative injury by up-regulating the expression of antioxidant enzymes. The C-PC treated astrocytes under this oxidative environment, also showed a significant increase in the mRNA levels of NGF and BNDF, two important neurotrophic factors with regenerative properties against hypoxic ischemic brain injury [106,107]. In this sense, it has been observed that astrocyte-derived BDNF supports OPCs maturation under hypoxic conditions *in vitro* and also oligodendrogenesis after white matter damage *in vivo*. Miyamoto et al. 2015 reported that OPCs maturation was suppressed when subjected to prolonged hypoxic stress by exposure to sublethal CoCl_2_. In contrast, when OPCs were incubated in an astrocyte conditioned medium, they were successfully differentiated into MBP^+^-labeled ODs even under hypoxic stress conditions, an effect that was significantly prevented when BDNF was removed by filtering from the astro-medium. The treatment with recombinant BDNF indeed promoted OPCs maturation under similar conditions. Transgenic mice (GFAP^cre^/BDNF^wt/fl^) subjected to prolonged cerebral hypoperfusion for 28 days, in which BDNF expression was downregulated specifically in astrocytes, showed a smaller number of mature ODs and also a smaller number of newly generated ODs in their corpus callosum as compared with wild-type mice, along with a decrease in myelin density [108].Taken together, this evidence suggests a positive role of C-PC in myelin regeneration after cerebral ischemia, by protecting astrocyte survival and inducing the BDNF expression in these cells, which then activate repair processes and accelerate endogenous oligodendrogenesis to alleviate white matter damage.

## 6. Conclusions

In summary, in the current review we present preclinical evidence indicating a remyelinating effect of C-PC and its tetrapyrrolic linked compound, PCB, in models of MS and IS. We discussed previous studies from our group showing that C-PC, given in therapeutic regimens, promoted white matter regeneration in the CNS of rats and mice affected by EAE. In particular, we presented new quantitative evidence of myelin formation in these models induced by C-PC, as assessed by the well-known *g*-ratio parameter in electron microscopy studies. Similarly, the *g*-ratio assessment of PCB effects in cerebral cortex samples from ischemic rats revealed a potent remyelinating action of this compound after the stroke. Among the multiple molecular mediators involved in such effects, we discussed the role of Treg induction, the control of pro-oxidant chemical species, the limitation of deleterious pro-inflammatory mediators, the modulation of gene expression, the inhibition of COX-2 and the induction of BDNF, as well as other biological processes that could be responsible for the actions of C-PC and PCB on the recruitment, differentiation and maturation of OPCs in demyelinated lesions present in MS and IS. The accumulated evidence supports the clinical application of these compounds as remyelinating therapies for enhancing the patient recovery in neurological diseases having a high disability incidence such as MS and IS.

## Figures and Tables

**Figure 1 behavsci-08-00015-f001:**
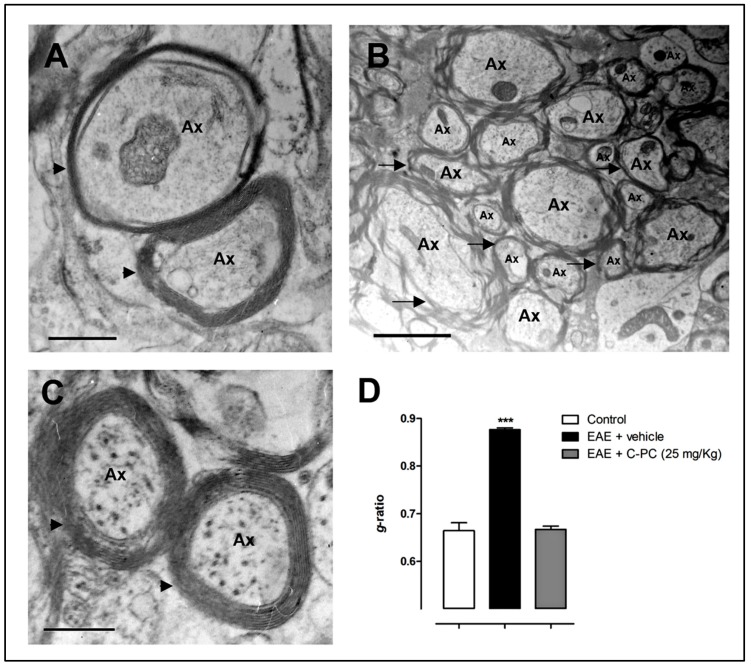
C-PC-induced remyelination in an EAE model in Lewis rats. Panels show representative electron microscopy images of brain biopsies from (**A**) non-diseased; (**B**) EAE-vehicle treated and (**C**) EAE-C-PC 25 mg/Kg treated animals; (**D**) *g*-ratio = [axon diameter/(myelin + axon) diameter] (mean ± S.E.M. *** *p* < 0.001 vs. control group, ANOVA + Newman-Keuls tests by using GraphPad Prism Software). C-PC at 25 mg/Kg or the vehicle were administered i.p. once a day from day 0 until day 12 after immunization (*n* = 10 each group). Control (non-immunized) animals received saline instead of encephalitogen. Brain samples were taken at the end of the study period (day 24 post-immunization). Arrowheads or arrows indicate normal myelin structure/thickness or myelin damage/loss, respectively. Ax: axon. Bar = 200 nm.

**Figure 2 behavsci-08-00015-f002:**
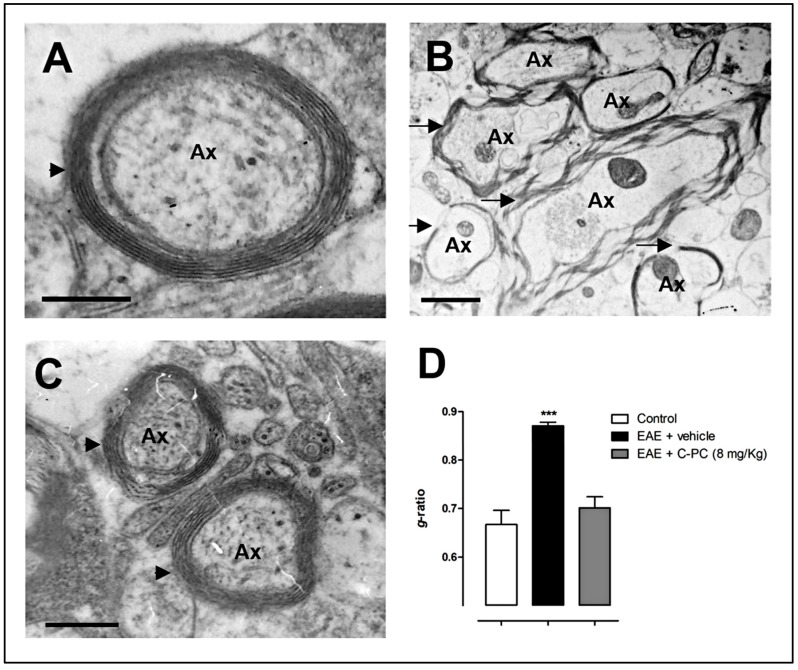
C-PC-induced remyelination in the MOG_35–55_-induced EAE model in C57BL/6 mice. Panels show representative electron microscopy images of spinal cord samples from (**A**) non-diseased; (**B**) EAE-vehicle treated and (**C**) EAE-C-PC 8 mg/Kg treated animals; (**D**) *g*-ratio = [axon diameter/(myelin + axon) diameter] (mean ± S.E.M. *** *p* < 0.001 vs. control group, ANOVA + Newman-Keuls tests using GraphPad Prism Software). C-PC at 8 mg/Kg or the vehicle were administered i.p. daily at disease onset (between days 11 and 12 post-immunization) for 15 days (*n* = 4–7 each group). Control (non-immunized) animals received phosphate buffered saline instead of encephalitogen. Spinal cord samples were taken at the end of the study period (day 29 post-immunization). Arrowheads or arrows indicate normal myelin structure/thickness or myelin damage/loss, respectively. Ax: axon. Bar = 200 nm.

**Figure 3 behavsci-08-00015-f003:**
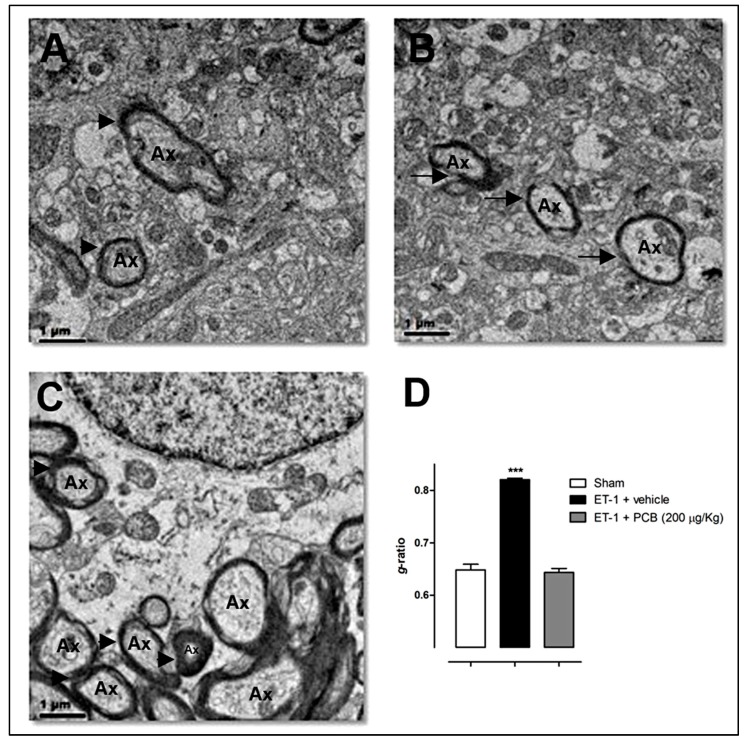
PCB-induced remyelination in a focal transient cerebral ischemia model induced by endothelin-1 (ET-1) in Wistar rats. Panels show representative electron microscopy images of cerebral cortex samples from (**A**) sham; (**B**) ET-1-vehicle treated and (**C**) ET-1-PCB 200 µg/Kg treated animals. (**D**) *g*-ratio = [axon diameter/(myelin + axon) diameter] (mean ± S.E.M. *** *p* < 0.001 vs. control group, ANOVA + Newman-Keuls tests by using GraphPad Prism Software). ET-1 at 600 pmol was stereotaxically injected in the piriform cerebral cortex at the coordinates +0.9 mm (anterior), +5.2 mm (lateral), −8.3 mm (dorsoventral) respective to bregma, adjacent to the middle cerebral artery. PCB 200 µg/Kg or the vehicle, were administered i.p. in equal subdoses at 30 min, 1, 3 and 6 h post-surgery (*n* = 5 each group). Sham animals received saline instead of ET-1. Brain samples were taken 24 h after the ischemic insult. Arrowheads or arrows indicate normal myelin structure/thickness or myelin damage/loss, respectively. Ax: axon. Bar = 1 µm.

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
