# Peer review of "C-Phycocyanin and Phycocyanobilin as Remyelination Therapies for Enhancing Recovery in Multiple Sclerosis and Ischemic Stroke: A Preclinical Perspective"

_behavsci, 2018, doi:10.3390/bs8010015_

Round 1

Reviewer 1 Report

The study investigates effect of C- Phycocyanin and Phycocyanobilin on regeneration of white matter affected by experimental autoimmune encephalomyelitis. The electron microscopy revealed a potent remyelinating action of PCB treatment after stroke. The authors proposed that the C-PC and PCB may play a role in regulation of T cell induction, the control of oxidative stress and pro-inflammatory mediators, gene expression modulation and COX-2 inhibition.                               

Current study is well written and presents valuable information on current topic. I have only few comments.

- Ln. 129. Please provide the reference for your report if available.

- The authors discuss their own data and information from other publications. The distinction is not always clear.

Author Response

Reviewer # 1

The study investigates effect of C- Phycocyanin and Phycocyanobilin on regeneration of white matter affected by experimental autoimmune encephalomyelitis. The electron microscopy revealed a potent remyelinating action of PCB treatment after stroke. The authors proposed that the C-PC and PCB may play a role in regulation of T cell induction, the control of oxidative stress and pro-inflammatory mediators, gene expression modulation and COX-2 inhibition.                              

Current study is well written and presents valuable information on current topic. I have only few comments.

1.       Ln. 129. Please provide the reference for your report if available.

Answer: A reference was included in Ln. 129. Reference # 34:

Boulanger, J.J.; Messier, C. From precursors to myelinating oligodendrocytes: contribution of intrinsic and extrinsic factors to white matter plasticity in the adult brain. Neuroscience 2014, 269, 343-366. doi: 10.1016/j.neuroscience.2014.03.063

2.       The authors discuss their own data and information from other publications. The distinction is not always clear.

Answer: The data obtained from our group was better described for improving the clarity and the distinction from the evidence reported by other groups included in the review. Changes were highlighted in yellow.

Reviewer 2 Report

Your paper is well-presented and despite its ambitiously wide field of study provides an interesting and compelling evidence for a potential molecular path to follow inducing remyelination in some CNS disorders. You mention (page 2, lines 69 and 70; reference 10) the numbers invoked in the Editorial you made reference to about 80% of RRMS patients evolve within 20 years into the SPMS form. While this happens in untreated RRMS as the natural (spontaneous) course of the disease, perhaps due to the nature of the informational message you are sending (modification of the course of diseases by repairing white matter elements), for your future information, there are reported clinical observations demonstrating that effective treatment of RRMS changes this proportion substantially. See Trojano et al, New Natural History of INF-β Treated Relapsing MS. Ann Neurol 2007: only 25% of treated RRMS converted to SP after 11.7 years of therapy. Also see Cree et al, Long-Term Evolution of MS Disability in the Treatment Era: 19% of treated individuals with any of the injectable platform therapies licensed from 1993-2001 transformed to SP after 16.8 years.

Your manuscript is well- designed, clearly presented and your figures arequite adequate.      

Author Response

Reviewer # 2

Your paper is well-presented and despite its ambitiously wide field of study provides an interesting and compelling evidence for a potential molecular path to follow inducing remyelination in some CNS disorders. You mention (page 2, lines 69 and 70; reference 10) the numbers invoked in the Editorial you made reference to about 80% of RRMS patients evolve within 20 years into the SPMS form. While this happens in untreated RRMS as the natural (spontaneous) course of the disease, perhaps due to the nature of the informational message you are sending (modification of the course of diseases by repairing white matter elements), for your future information, there are reported clinical observations demonstrating that effective treatment of RRMS changes this proportion substantially. See Trojano et al, New Natural History of INF-β Treated Relapsing MS. Ann Neurol 2007: only 25% of treated RRMS converted to SP after 11.7 years of therapy. Also see Cree et al, Long-Term Evolution of MS Disability in the Treatment Era: 19% of treated individuals with any of the injectable platform therapies licensed from 1993-2001 transformed to SP after 16.8 years.

Your manuscript is well- designed, clearly presented and your figures are quite adequate.     

Answer: Authors agree with the reviewer´s comment and are thankful for this information. The sentence was corrected accordingly (page 2, lanes 69-72).
